# Genomic discovery of EF-24 targets unveils antitumorigenic mechanisms in leukemia cells

**Ajeet P. Singh***, **Noah Wax, James Duncan, Ana S. Fernandes, Jonathan L. Jacobs**\*

American Type Culture Collection (ATCC), Manassas, Virginia, United States of America

\* asingh@atcc.org (APS); jjacobs@atcc.org (JLJ)

## Abstract

Curcumin, a polyphenolic compound derived from the plant *Curcuma longa* L., has demonstrated a wide range of therapeutic properties, including potential anti-cancer effects. However, its clinical efficacy is limited due to poor bioavailability and stability. To overcome these challenges, curcumin analogs like EF-24 have been developed with improved pharmacological properties. In this study, in order to improve our understanding of EF-24's potential mechanisms of action, we used whole-transcriptome sequencing to identify genome-wide functional impacts of EF-24 treatment in leukemia cells. These results enabled the development of a testable model system for associating druggable genes with clinical disease targets related to EF-24 treatment. To develop our model of the transcriptional response to EF-24 treatment, we used four well studied model cell lines for leukemia research, specifically the chronic myeloid leukemia (CML) cell line K-562 and acute myeloid leukemia (AML) cell lines HL-60, Kasumi-1, and THP-1. Cell viability was significantly decreased in all four of these leukemia models following EF-24 treatment as compared to untreated controls. We discovered that the genes ATF3, CLU, HSPA6, OSGIN1, ZFAND2A, and CXCL8, which are associated with reduced cell viability and proliferation, were consistently upregulated in all EF-24–treated cell lines. Further analysis of the tested cell lines revealed the activation of various signaling pathways, but notably the S100 family signaling pathway was consistently activated in all four cell lines. Our results provide critical insights into the molecular underpinnings of EF-24's antitumor efficacy against leukemia subtypes, highlighting its multifaceted impact on signaling pathways and gene networks that regulate cell survival, proliferation, and immune responses in cell line models of myeloid leukemia subtypes.

## Introduction

Chronic (CML) and acute myeloid leukemia (AML) are heterogeneous hematological malignancies that arise from the rapid, uncontrolled proliferation of immature or abnormal myeloid blood cells [1]. The complexity of these diseases is further

**Data availability statement:** All RNAseq data produced for this study is available under NCBI Bioproject PRJNA1165739.

**Funding:** The author(s) received no specific funding for this work.

**Competing interests:** The authors have declared that no competing interests exist.

**Abbreviations:** AML, Acute myeloid leukemia; ATCC, American Type Culture Collection; CML, Chronic myeloid leukemia; DEG, Differentially expressed genes; FBS, fetal bovine serum; IMDM, Iscove's Modified Dulbecco's Medium; PCA, Principal component analysis; RNAseq, RNA sequencing; RPMI, Roswell Park Memorial Institute Medium.

exacerbated by diverse subtypes, each characterized by distinct genetic alterations and clinical behaviors. While traditional chemotherapy has improved survival rates for many leukemia patients, its efficacy is often limited by drug resistance, off-target toxicity, and disease relapse [2]. This underscores the urgent need for more selective and less toxic therapeutic alternatives.

Curcumin, a bioactive polyphenolic compound extracted from the rhizomes of *Curcuma longa* L., has garnered significant attention for its potential anticancer properties [3]. Despite its promising effects in preclinical studies, curcumin's clinical translation has been hindered by its poor bioavailability and rapid metabolism. To overcome these limitations, synthetic curcumin analogs with enhanced pharmacokinetic properties and greater potency have been developed [4]. Among these, EF-24 (diphenyl difluoroketone) has emerged as a promising anticancer agent due to its potent antiproliferative and pro-apoptotic effects across multiple cancer models [3]. EF-24 exerts its anticancer activity by modulating key signaling pathways implicated in cell cycle regulation, apoptosis, and inflammation.[5] Notably, it has been shown to induce oxidative stress, disrupt NF-κB signaling, and trigger mitochondrial-mediated apoptosis in various cancer types.[6] Additionally, EF-24 has been reported to inhibit PI3K/Akt and STAT3 pathways, which are frequently dysregulated in leukemia and contribute to cell survival and chemoresistance.[7] Despite these advances, the precise molecular targets and genome-wide transcriptional effects of EF-24 in leukemia subtypes remain poorly understood [5,6].

To elucidate EF-24's mechanisms of action in leukemia, we investigated its effects on a panel of well-characterized CML (K-562 (ATCC® CCL-243™) and AML HL-60 (ATCC® CCL-240™), Kasumi-1 (ATCC® CRL-2724™), and THP-1 (ATCC® TIB-202™) cell lines. The CML cell line K-562 (ATCC® CCL-243™) was selected due to its BCR-ABL fusion gene, which drives leukemogenesis and is a key therapeutic target in CML [7]. The AML cell lines HL-60 (ATCC® CCL-240™) [8], Kasumi-1 (ATCC® CRL-2724™), and THP-1 (ATCC® TIB-202™) were chosen to represent distinct AML subtypes, each with unique genetic and phenotypic features. HL-60 cells model early-stage AML and are widely used for studying myeloid differentiation and chemotherapeutic responses [9]. Kasumi-1 cells harbor the *t(8;21)* translocation, a hallmark of core-binding factor AML, which is associated with specific transcriptional dysregulation [9]. THP-1 cells, derived from an M5 monocytic AML subtype, provide a model for investigating leukemic monocyte-macrophage differentiation and inflammatory signaling [9].

These cell lines represent specific myeloid leukemia subtypes and have been extensively characterized and widely used in research, offering unique insights into the pathogenesis and therapeutic vulnerabilities of the disease. Here, our objective was to first illuminate the baseline transcriptional profiles of these leukemia cell lines, emphasizing the existing molecular discrepancies among them, then conduct RNA-sequencing to explore global gene expression landscape in responses to EF-24 exposure. Leveraging comparative differential gene expression analysis of untreated and EF-24–treated cells, we uncovered the molecular pathways and gene regulatory networks underlying EF-24's anticancer effects. Overall, this approach provided an

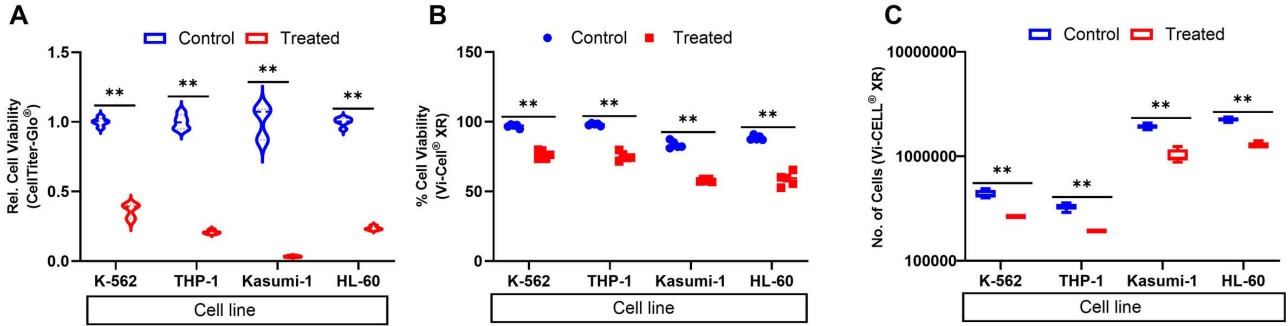

unbiased assessment of global gene expression and target gene relative abundance, enabling the identification of differences and similarities among each cell line in response to EF-24 treatment.

## Results

### EF-24 exhibits potent cell killing activity in leukemia cell lines

We evaluated the antitumor efficacy of EF-24 across four genetically distinct leukemia cell lines—K-562, Kasumi-1, THP-1, and HL-60—selected to represent key subtypes of myeloid leukemia [10–14]. Building on the foundational work by Hsiao et al., which demonstrated EF-24-induced cytotoxicity in AML cell lines, we sought to validate and extend those findings [5]. Their study reported elevated levels of cleaved caspase-3 and PARP following treatment with 2 µM EF-24, indicative of pronounced apoptotic cell death [5]. To assess EF-24's cytotoxic potential, we treated each cell line with 2 µM EF-24 and measured viability using the CellTiter-Glo® assay (Promega®) 24 hours post-treatment. The results revealed substantial reductions in cell viability: approximately 60% in K-562, 75% in THP-1 and HL-60, and 90% in Kasumi-1 (Fig 1A). These findings were corroborated by direct cell counts, which consistently showed decreased numbers of viable cells in treated samples compared to untreated controls (Fig 1B,1C). Collectively, these data reinforce previous findings and highlight EF-24's potent cytotoxic activity across diverse myeloid leukemia subtypes, supporting its continued evaluation as a promising therapeutic candidate with broad applicability across lineage-specific, hematologic and solid tumor models, including translational relevance in primary patient samples [15].

### Chronic (CML) and Acute Myeloid Leukemia (AML) cell lines exhibit transcriptional heterogeneity

To understand the inherent molecular characteristics of the myeloid leukemia cell lines, we carried out whole-transcriptome profiling to establish a reference baseline for gene expression in naive cells. Principal component analysis (PCA) revealed close clustering of biological replicates (n = 5), while leukemia cell lines were stratified into four distinct clusters based on their transcript and gene expression levels, indicating that each of these cell lines represents a distinct model system and transcriptional "starting point" for studying the impact of drug (EF-24) treatment response. Comparative gene expression analysis between the cell lines revealed differential patterns in the top 100 genes, providing insights into their roles in specific cell types (Fig 2A). This facilitated the identification of marker genes specific to each leukemia cell line. For example, *HBG1*, *RHAG*, *HBG2*, *HBA1*, and *NMU* are enriched in K-562 cells as compared to THP-1, Kasumi-1, and HL-60. In contrast, *SMAP2*, *ZMPSTE24*, *CDC20*, *NFYC*, and *RPS4Y1* are enriched in THP-1. Similarly, *CD34*, *KIT*, *EEF1A1*, *PRSS57*, *RPLP0*, and *RPS24* are enriched in Kasumi-1, while *MT-ND4L*, *MT-ND4*, *MT-CO1*,

**Fig 1. EF-24 treatment significantly reduces cell viability in leukemia cell lines.** (A) Relative cell viability measured using the Promega® CellTiter-Glo® luminescent assay shows a marked decrease in EF-24–treated cells compared to controls. (B) Percent viability and (C) total cell counts assessed using the Beckman Coulter® Vi-CELL® XR analyzer further confirm the cytotoxic effect of EF-24, demonstrating consistent reductions in both viability and cell number across treatment groups. Statistical p- value ** < 0.01.

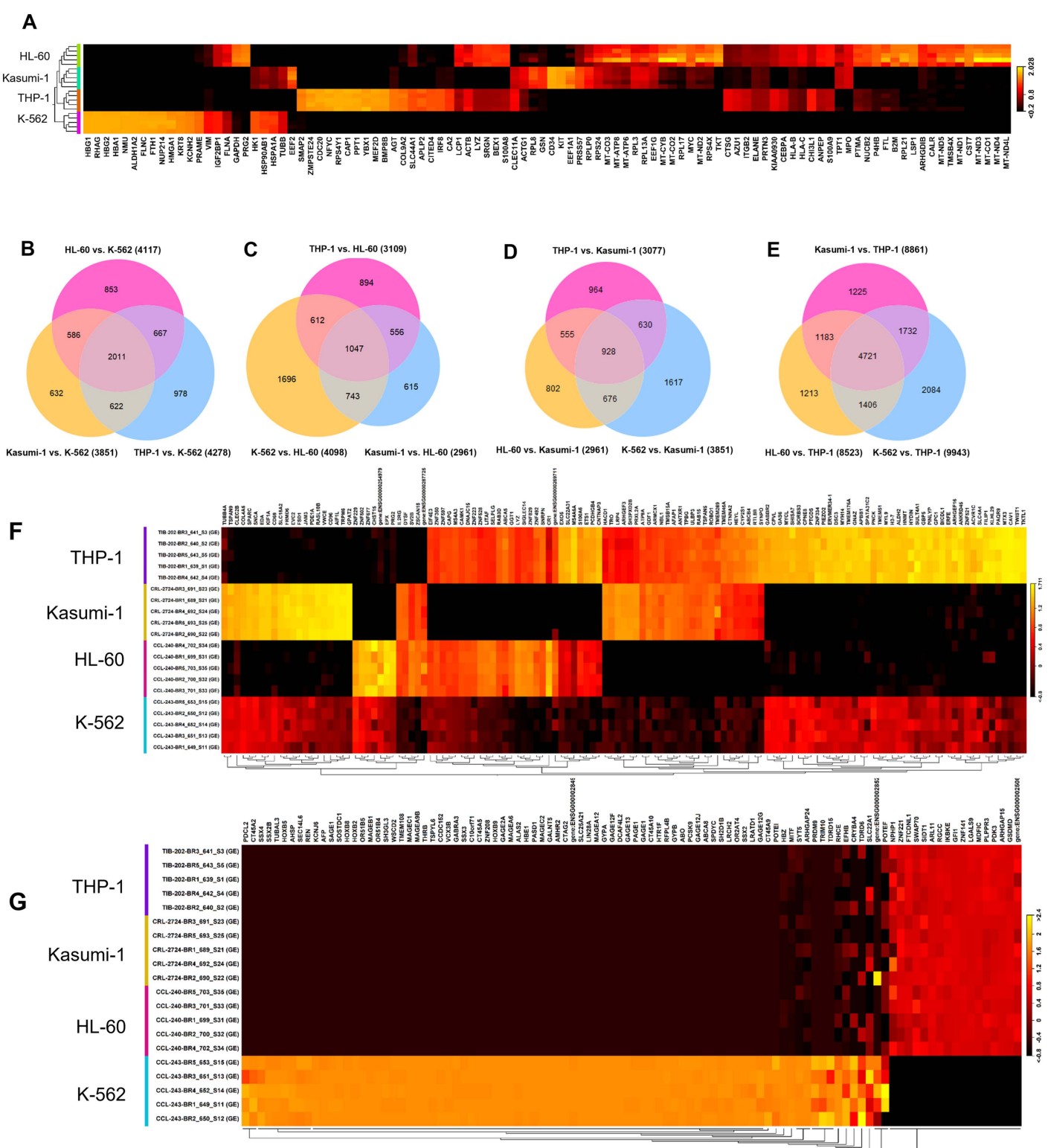

**Fig 2. Distinct and shared gene expression patterns among leukemia cell lines following EF-24 treatment.** (A) Heatmap demonstrates the clustering pattern of genes differentially expressed among leukemia cell lines (red = induced; black = reduced). Each cell line has 5 biological replicates. (B) Venn diagram showing the shared and unique genes differentially expressed in the AML subtypes HL-60, THP-1, and Kasumi-1 as compared to the CML subtype K-562. (C) Venn diagram showing the shared and uniquely expressed genes in K-562, THP-1, and Kasumi-1 as compared to HL-60. (D) Venn

diagram showing the shared and uniquely expressed genes in THP-1, K-562, and HL-60 that differ from those in Kasumi-1. (E) Venn diagram showing the shared and uniquely expressed genes in Kasumi-1, HL-60, and K-562 that differ from those in THP-1. For each Venn diagram, the differences in circle size reflect the number of genes. (F) The heatmap displays the DEGs genes exhibiting inconsistent expression patterns in THP-1, Kasumi-1, and HL-60 as compared to K-562. (G) The heatmap displays the genes exhibiting a consistent expression pattern in the THP-1, Kasumi-1, and HL-60 as compared to K-562. Absolute fold change ≥5, false discovery rate (FDR) p-value <0.01.

*MT-ND3*, and *CST7* are enriched in HL-60 cells (Fig 2A). The expression patterns of these genes in particular cell lines could serve as biomarkers for characterizing specific leukemia subtypes and as potential pharmacological targets.

When further analyzing the number of differentially expressed genes (DEGs) either shared or unique in the AML cell lines as compared to the CML cell line, we identified 2011 genes that were found to be differentially regulated and shared across the THP-1, Kasumi-1, and HL-60 cell lines and distinct from the K-562 cell line (Fig 2B). We also discovered numerous genes unique to HL-60 (853), THP-1 (978), and Kasumi-1 (632) as compared to K-562 (Fig 2B,2F). Further investigation of HL-60, Kasumi-1, and THP-1 revealed genes that shared similar expression patterns among multiple cell lines as well as those unique to specific cell lines (Fig 2C,2D,2E,2F). We then looked for the genes shared by THP-1, Kasumi-1, and HL-60 that could be a common factor driving the AML subtypes; here, we identified genes with consistent expression patterns among the cell lines. For instance, *PDCL2*, *CT45A2*, *SSX4*, *SSX2B*, *TUBAL3*, *HOXB5*, *AHSP*, and *SEC14L6* are consistently downregulated whereas *GSDMD*, *ARHGAP15*, *PDK3*, *PLPPR3*, *MDFIC*, *LGALS9*, *ZNF141*, and *GFI1* are upregulated in HL-60, Kasumi-1, and THP-1 as compared to K-562 (Fig 2G). Next, we identified genes inconsistently regulated in THP-1, Kasumi-1, and HL-60 cells and differentially expressed as compared to K-562 cells. For example, *TUBB4A4*, *TSPAN9*, *CLEC2B*, *COL4A5*, *SPARC*, and *SNCA* are highly augmented in Kasumi-1 cells but attenuated in HL-60 and THP-1 cells as compared to K-562 cells (Fig 2F). Similarly, we also identified a subset of genes differentially enriched or decreased in HL-60 (e.g., *TUBB4A*, *TSPAN9*, *CLEC2B*, *COL4A5*, *SPARC*, *SNCA* decreased; *ZNF229*, *ZNF502*, *ZNF67*, *CHST15*, *EPX* increased) and THP-1 (e.g., *IL2RG*, *DYSF*, *SV2B*, *ZSCAN18* decreased; *TKTL1*, *TWIST1*, *CAV1*, *MTX3*, *PAQR9*, *KLHL29*, *FILIP1*, *SLC4A4*, *ACVR1C* increased) as compared to K-562 (Fig 2F). This comprehensive analysis revealed the global molecular heterogeneity between cell lines and marker genes, underscoring the importance of evaluating different subtypes when reviewing a new treatment as it may affect each cell line differently.

## EF-24 treatment impacts the expression of genes controlling cell survival and proliferation

To identify the gene expression changes responsible for EF-24's anticancer effects in leukemia cell lines, we performed RNAseq on EF-24-treated cells and compared them against untreated baseline controls. Principal Component Analysis (PCA) carried out RNAseq profiles for EF-24 treated and untreated samples, with all biological replicates of each condition clustering together (Fig 3A). Furthermore, differential gene expression analysis highlighted a shift in the pattern of gene expression in EF-24–treated cells as compared to control cells (Fig 3B).

Regardless of the distinct myeloid leukemia subtypes, subsequent analysis uncovered 63 differentially expressed genes shared among all four leukemia cell lines treated with EF-24 as compared to the untreated controls of each cell line (Fig 3C). The number of genes altered in individual cell lines following EF-24 treatment was identified (Fig 3C). Several genes known to inhibit cell viability (e.g., *HSPA6*, *CLU*, *OSGIN1*, *ATF3*, *CXCL8*, *ZFAND2A*, and *S100A10*) exhibited a multifold increase in expression level in all four EF-24–treated cell lines as compared to the respective untreated control for each cell line (Fig 3D,3E). Further, Ingenuity Pathway Analysis (IPA) of genes that were both differentially expressed and shared in all four EF-24–treated leukemia cell lines revealed the top canonical pathways, including the S100 family signaling pathway (Fig 3F). The activation of S100 signaling causes the induction of downstream effector molecules such as *JAK1*, *TYK2*, and *P38 MAPK*, which induce NFkB-regulated proinflammatory cytokines linked to cell proliferation, survival, and viability [16].

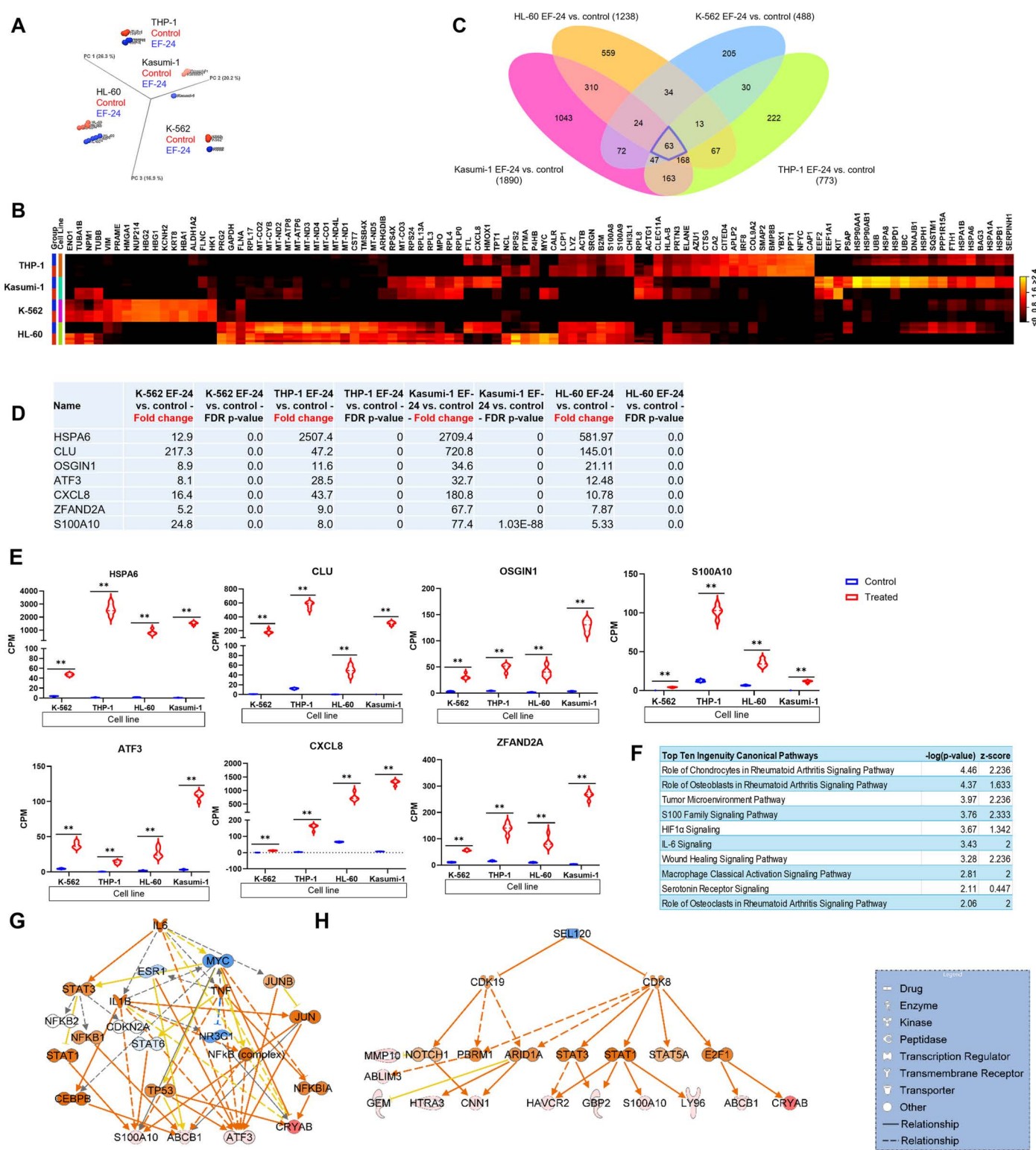

**Fig 3. EF-24 treatment disrupts key molecular pathways associated with cell survival, proliferation, and viability.** (A) PCA plot illustrates sample clustering in leukemia cell lines and experimental groups. The variations within the samples resulted in distinct clustering (red = control samples; black = EF-24 samples). (B) Heatmap of the top 100 DEGs in leukemia cell lines that shows the highest fold change in EF-24–treated cells as compared

to untreated controls. There are 5 biological replicates of each condition/cell line (red = control; blue = treated). (C) Venn diagram depicts shared and unique genes that are differentially expressed in EF-24–treated cells as compared to untreated controls. Threshold absolute fold change ≥5, FDR p-value 0.01. (D) Pan differentially expressed genes in EF-24–treated cells as compared to untreated controls. (E) Violin plots display the relevant genes' quantitative enrichment in EF-24–treated cells as compared to untreated controls. (F) IPA shows shared top canonical pathways activated in myeloid leukemia cell lines were treated with EF-24. (G) IL-6 is an upstream regulator of genes induced in the EF-24 cell lines. (H) Regulatory network of genes induced in the EF-24 treated cell lines. False Discovery Rate (FDR) p-value **< 0.01.

We then investigated upstream regulators of genes differentially expressed in EF-24-treated cells. IL-6 emerged as a key regulator, activating *JUN*, *NFkB*, and *STAT1* that causing induction of downstream genes S100A10, ABCB1, ATF3 and CRYAB adversely impacting cell viability and survival (Fig 3G). Moreover, NUPR1, appeared to be a novel transcription regulator, upregulated in response to EF-24 treatment and directly control expression of induced genes such as *ATF3, CXCL8, DUSP8, MMP10, UPP1, ZFAND2A,* AND *ADM.* Additionally, networks of upregulated genes in EF-24–treated cells suggest associations with cell death and survival (Fig 3H). Overall, EF-24 treatment causes adverse effects on myeloid leukemia cell lines through alterations in key pathways and a highly interconnected network of genes that negatively impact cell survival, proliferation, and viability. To further understand the transcriptional response to EF-24 therapy in specific leukemia subtypes, we explored differentially expressed genes and pathways at the individual cell line level.

## EF-24 treatment activated the wound healing signaling pathway in K-562 cells

To identify the specific genes and pathways in K-562 cells affected by EF-24, we evaluated the differentially expressed genes. The expression patterns in heatmap were color-coded: red indicated increased expression, while black indicated decreased expression in treated cells compared to untreated control cells (Fig 4A). Among the differentially regulated genes, 488 genes exhibited a ≥ 5-fold change (p-value 0.01) in EF-24–treated cells as compared to untreated K-562 controls; only a small subset of DEGs (9 out of 488 genes) were downregulated. We then focused on 25 upregulated genes that showed the highest fold change in the list of DEGs (Fig 4B). Interestingly, *CLU, PTPRN, NDRG1, GBP2, OSGIN1, ATF3, IFI16, HLA-C, BEX2,* and *VWA5A*—known for their tumor suppressive function—were induced in many folds (Fig 4C) [17]. Count-per-million (CPM) values are used to filter genes in RNA-seq analysis with a minimum cut-off of 0.5 CPM. Gene falling below this threshold across all samples were excluded from further analysis.

Next, analysis in IPA unveiled canonical pathways and gene regulatory networks of the genes activated in response to EF-24 treatment, shedding light on EF-24 mechanisms influencing cell viability. In this pursuit, IPA illuminated the wound healing signaling pathways and their downstream effector molecules that potentially orchestrate gene expression changes in EF-24–treated cells (Fig 4D). This activation triggered MAP3K7-mediated NFkB and its target genes, which are crucial players in controlling cell survival and death [18]. Additionally, within the wound healing signaling pathway, the EGFR signaling cascade was activated, leading to the activation of downstream genes like *RAS, MEK,* and *ERK1/2*. These, in turn, transactivated *AP1* transcription factor–regulated suppression of *clusterin* (*CLU*) gene that contributes to tissue repair [19].

Advancing in our transcriptional analysis, we uncovered upstream molecules governing gene expression in EF-24–treated cells (Fig 4E,4F). The activation of proinflammatory cytokines IL-1B and TGFB1 modulates downstream genes such as *NFKB1, RELA, TNF, JUN, TP53, CEBPB,* and *FOS*; each of these functions as tumor suppressors (Fig 4E). Meanwhile, the induced expression of various genes, including *STAT3, SP1,* and TNF is indirectly promoted and impeded by anti-inflammatory dexamethasone respectively (Fig 4F), which directly activates the repressed NR3C1 glucocorticoid receptor (GR) [20]. This comprehensive analysis not only enhances our understanding of the molecular responses triggered by EF-24 but also underscores the intricate interplay of pathways and regulatory networks contributing to the observed cellular effects in K-562 cells.

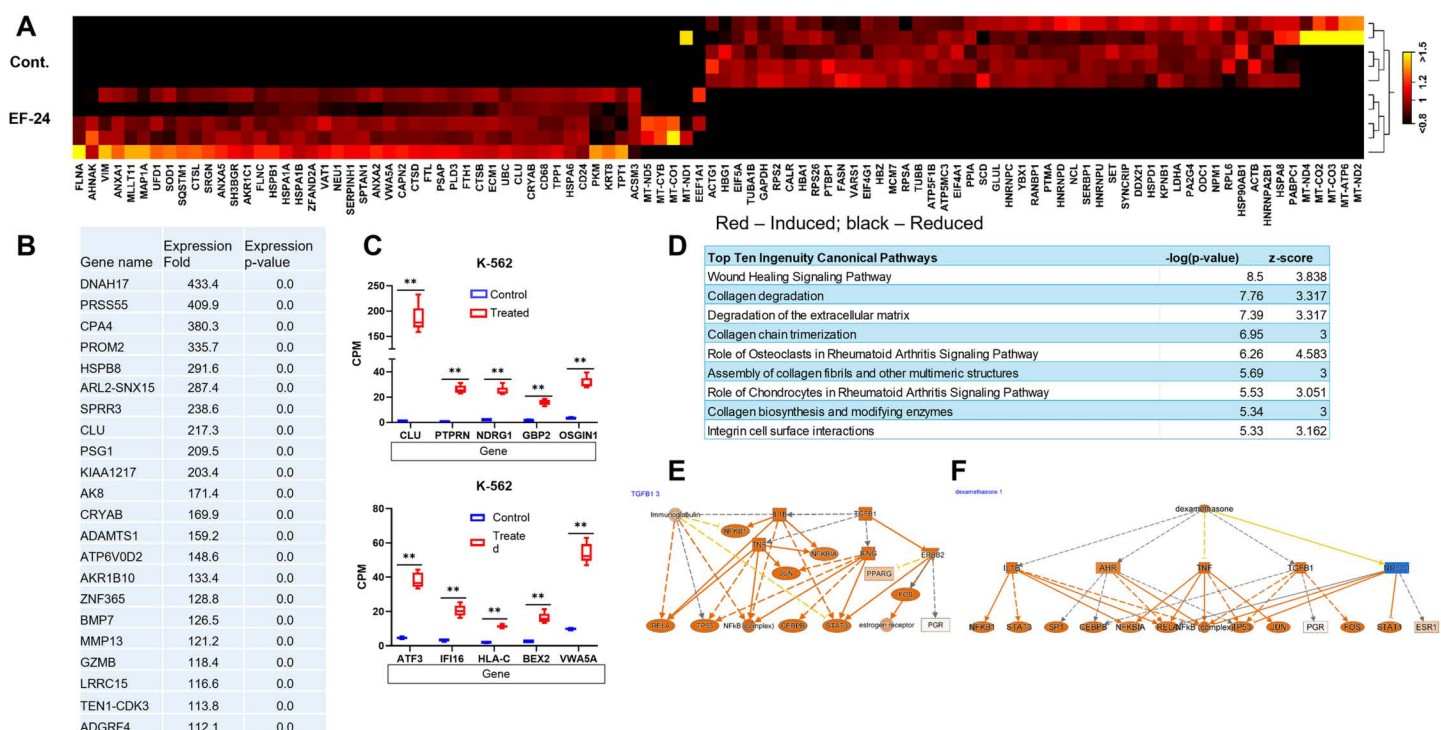

**Fig 4. EF-24 induces significant transcriptional changes in K-562 cells.** (A) Heatmap of the top 100 genes differentially regulated in EF-24–treated cells versus the untreated control (red = induced; black = reduced). Cutoff fold change 5 ≥ p-value <0.01. (B) Top 25 molecules changed in EF-24–treated cells versus untreated control K-562 cells. (C) Box plots show relative mRNA expression level of indicated genes in EF-24–treated and untreated control cells. ≥ 5, FDR p-value 0.01. (D) IPA shows shared top canonical pathways activated in K-562 cells that were treated with EF-24. (E, F) Network of genes predicted to be induced by IL1B and Dexamethasone in K-562 cells treated with EF-24. False Discovery Rate (FDR) p-value ** < 0.01.

## EF-24–activated pathways cause neutrophil degranulation in HL-60 cells

We performed full transcriptome analysis on HL-60 cells treated or untreated with EF-24 to understand transcriptional alterations and uncover signaling pathways and gene regulatory networks. Differential expression analysis revealed that EF-24 treatment had dramatically changed genes as compared to that of the untreated controls (Fig 5A). Quantitative analysis of DEGs showed that EF-24 treatment significantly increased gene expression levels (red color in heatmap) as compared to untreated controls.

To understand the functional implications of top fold changed DEGs, we employed gene ontology analysis with Gene-Cards. The analysis indicated that the genes *MAFA, CLU, RHOB, RASD1, HSPA1B, HSPA1A, HMOX1, CXCL8, DNAJB1, NCF2, SAT1, IER5, BAG3, B4GALNT1, CCL3, ANXA2, PPP1R15A, CES1*, and *JUN*, which were induced in EF-24–treated cells (Fig 5B), may participate in cancer progression [17,21–23]. However, it is noteworthy that their excessive expression level causes adverse effects such as the disruption of homeostasis and cell death [22,24,25]. This indicates that the overexpression of genes that promote tumor growth affected cellular homeostasis, resulting in a decrease in cell survival and an increase in cell death. Conversely, the genes *FTH1, SQSTM1, HSPA6, TSPYL2, PHLDA2, ITGAX*, and *DNAJA4* are expected to act as versatile protein regulators, potentially suppressing tumor cell genetic instability and promoting autophagy, apoptosis, and other forms of cell death [26–31].

We then focused on down-regulated genes associated with cancer, revealing several genes that play pivotal roles in various aspects of cancer development, including cell proliferation, adhesion, migration, phagocytosis, integrin signal

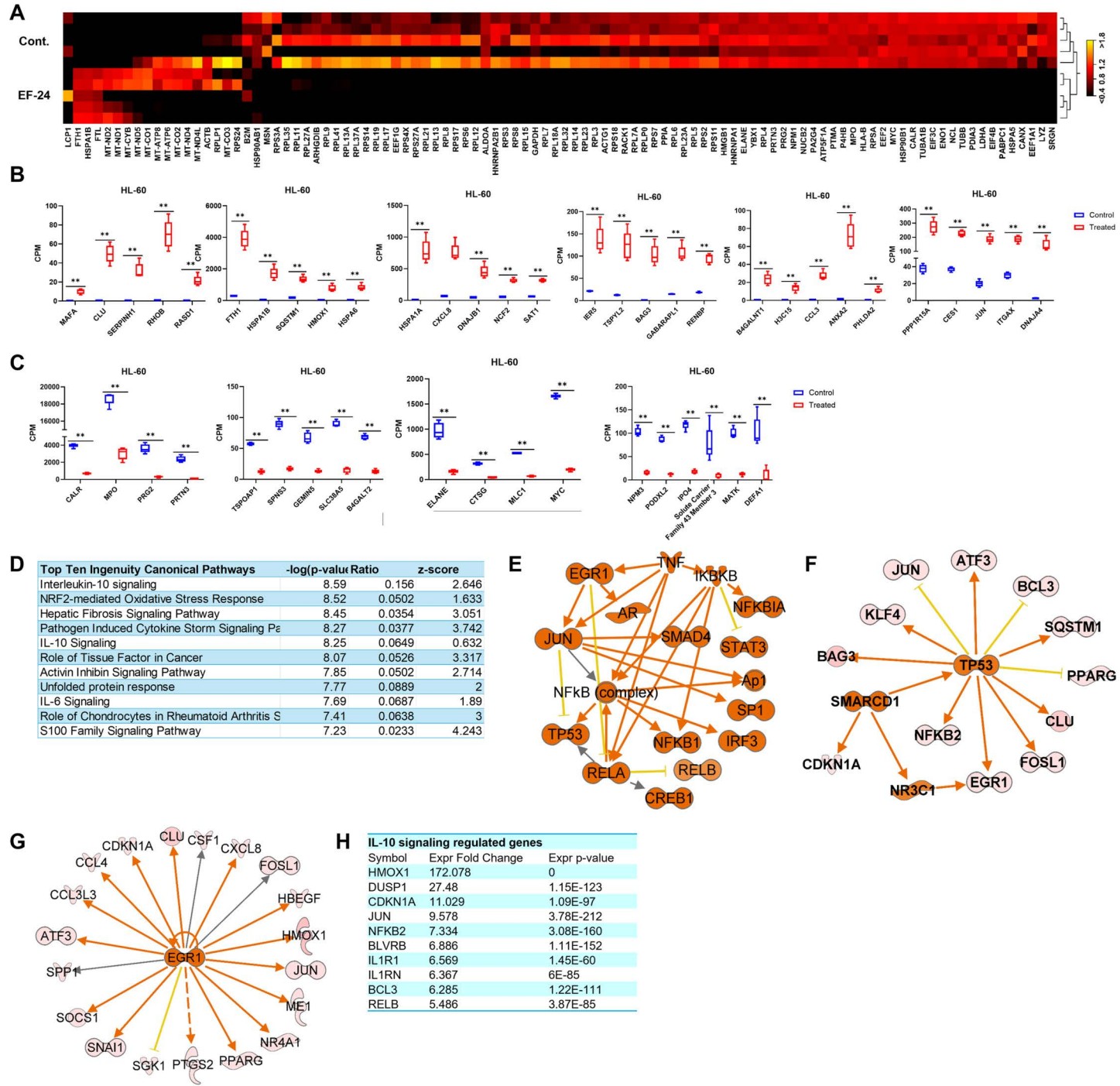

**Fig 5. EF-24 treatment alters gene expression profiles in HL-60 cells.** (A) Heatmap of the top 100 genes altered in HL-60 treated cells as compared to untreated control (red = induced; black = reduced). (B) Box plots show relative mRNA level of the indicated genes induced in EF-24–treated as compared to untreated control HL-60 cells. (C) Box plots show relative mRNA level of the indicated genes reduced in EF-24–treated as compared to untreated control HL-60 cells. Threshold absolute fold change ≥5, FDR p-value 0.01. (D) IPA shows top canonical pathways activated in HL-60 cells were treated with EF-24. (E) Gene network of genes regulated by TNF in HL-60 cells treated with EF-24. (F, G) Causal gene networks controlled by SMARCD1 and EGR1 in EF-24–treated HL-60 cells. (H) Bar plot shows relative mRNA level of the indicated downstream target genes of the IL-10 signaling in EF-24 treated as compared to untreated control HL-60 cells. False Discovery Rate (FDR) p-value ** < 0.01.

transduction, and immunogenic cell death (ICD) (Fig 5C). Interestingly, certain repressed genes like *B4GALT2*, *ELANE*, *CTSG*, *MLC1*, *NMP3*, and *SLC43A3* act as cancer suppressors, inhibiting malignant features and cancer development [32–34]. This suggests that EF-24 triggers a defense process under stressed physiological conditions that influences both protumorigenic and antitumorigenic gene expression.

Using IPA, we explored the pathways and upstream regulators of DEGs in EF-24–treated HL-60 cells and identified a downregulation of the neutrophil degranulation pathway. This downregulation contributes to an unfavorable environment for leukemia cell survival, leading to cell death [35]. On the other hand, several signaling pathways, with IL-10 ranking as the top upregulated pathway, were identified (Fig 5D). These findings demonstrate that EF-24 activates IL-10 signaling–associated transcriptional programs in leukemia cell lines, with both the direction and magnitude of gene expression changes supporting its engagement in tumor-suppressive pathways.

Notably, EF-24 treatment activated signaling cascades, including STAT3, that traditionally impede apoptosis, support cell proliferation, facilitate angiogenesis, and suppress antitumor immune responses [36]. This paradoxical activation of proinflammatory and cell survival pathways suggests a unique mechanism of EF-24-induced cell death in leukemia subtypes. TNF and dexamethasone modulated the expression of altered genes in EF-24 treated HL-60 cells and function as upstream regulators (Fig 5E). Causal network analysis in IPA highlighted SMARCD1 as an upstream regulator, playing a vital role in gene expression regulation through chromatin remodeling [37]. SMARCD1 activity directly transactivates *P53*, CDKN1a, and *NR3C1* in EF-24–treated HL-60 cells (Fig 5F), inhibiting cell viability. Additionally, EGR1 activation and its target genes were observed in EF-24–treated cells (Fig 5G). Given IL-10's context-dependent role in cancer—exhibiting both tumor-promoting and tumor-suppressive effects [38], we extend our analysis to explore IL-10 signaling associated genes in HL-60 cells. We performed a focused analysis of genes identified through IPA. EF-24 treatment was associated with upregulation of IL-10–responsive genes including HMOX1, DUSP1, CDKN1A, JUN, NFKB2, BLVRB, IL1R1, IL1RN, BCL3, and RELB (Fig 5H), all previously linked to cell death [39]. These findings suggest that EF-24–induced IL-10 activation contributes to cell death in HL-60 cells. Additional IL-10–associated genes such as CSF1, CXCL8, and CCL4 also showed significant expression changes (S1A Fig). Extending this analysis to K-562, Kasumi-1, and THP-1 cells revealed consistent modulation of IL-10 signaling–regulated genes across all four cell lines following EF-24 treatment (S1B–D Fig). Functional annotation revealed that the genes were activated associated with activation of leukocytes, differentiation of hematopoietic progenitor cells, and cell death of tumor. In summary, our comprehensive analysis of EF-24–treated HL-60 cells revealed a complex interplay of gene expression changes, signaling pathways, and regulatory networks contributing to cell death. The findings underscore the potential of EF-24 as a promising agent for leukemia treatment, warranting further investigation in preclinical and clinical settings.

**EF-24 activates tumor suppressor genes in Kasumi-1 cells**

To investigate the gene expression patterns in Kasumi-1 cells in response to EF-24, we conducted a differential gene expression analysis in treated versus untreated cells. Our findings revealed DEGs were either upregulated or downregulated (red-increase, dark-decrease) in EF-24–treated cells when compared to control cells (Fig 6A). The treatment with EF-24 induced a significant multifold change in the relative expression levels of various genes; notably, many of these induced genes, including *FTL*, *UBC*, *FTH1*, *UBB*, *SQSTM1*, *JUN*, *MLLT1*, *GADD45A*, *KLF6*, *PPP1R15A*, *TSPYL2*, *SRGN*, *SRXN1*, *NDRG1*, and *ZFP36*, function as tumor suppressors. In contrast, tumor promoters such as *MYBL2*, *TYMS*, *MYC*, *MCM7*, and *EGFL7* were downregulated in EF-24–treated cells (Fig 6B).

Pathway analysis further unveiled top dysregulated canonical pathways, including the Pathogen Induced Cytokine Storm Signaling (PICSS) Pathway and IL-10 signaling, which were not only activated in EF-24–treated Kasumi-1 cells (Fig 6C) but also in HL-60 cells. Excessive production of inflammatory signals by the immune system can result in a cytokine storm, potentially leading to cell death.[34] Moreover, TP53 and TP63, which function as tumor suppressors, appeared as upstream regulators of genes induced in EF-24–treated Kasumi-1 cells (Fig 6D).

Considering the downregulation of a substantial number of genes by EF-24, we investigated the pathways controlling these downregulated genes. Pathways crucial for cell cycle control, chromosomal replication, and DNA synthesis were

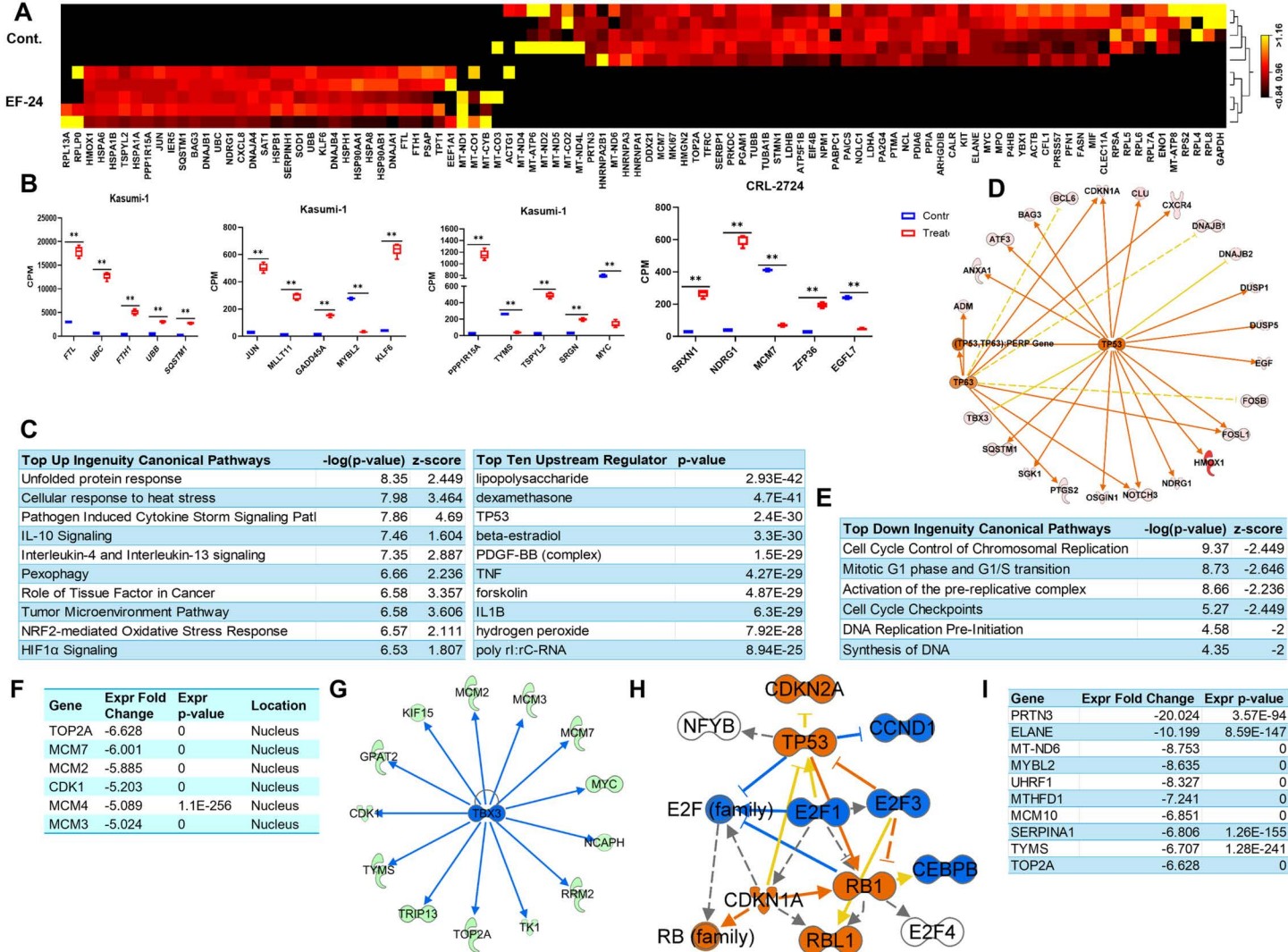

**Fig 6. EF-24 treatment induces distinct gene expression changes in Kasumi-1 cells.** (A) Heatmap of the top 100 genes differentially expressed in the EF-24–treated Kasumi-1 cells (red = induced; black = reduced). Absolute fold change >5, FDR p-value 0.01. (B) Box plots display the relative levels of candidate genes differentially expressed in the RNA-Seq datasets of EF-24–treated Kasumi-1 cells. (C) Top canonical pathways activated in EF-24–treated Kasumi-1 cells. (D) TP53 and TP63 are the upstream regulators of genes induced in the EF-24–treated Kasumi-1 cells. (E) Top canonical pathway down regulated in the EF-24–treated Kasumi-1 cells. (F) Cell cycle and chromosomal replication pathway associated with down regulated genes in the EF-24–treated Kasumi-1 cells. (G, H) Network of down regulated genes in the EF-24–treated Kasumi-1 cells regulated by TBX3 and E2F1. () Table of top-down regulated genes show highest fold change in EF-24–treated Kasumi-1 cells. False Discovery Rate (FDR) p-value ** < 0.01.

downregulated, leading to a multifold decrease in the expression levels of core molecules such as *TOP2A*, *MCM7*, *MCM2*, *CDK1*, *MCM4*, and *MCM3* (Fig 6E, 6F). Additionally, TBX3 and E2F, identified as upstream regulators, were inhibited in EF-24–treated cells, resulting in the downregulation of their downstream target genes (Fig 6G, 6H, 6I).

## The antitumorigenic effects of EF-24 in THP-1 cells are mediated by interferon alpha/beta signaling

Our analysis of differential gene expression in the comprehensive transcriptomic data from EF-24–treated and untreated THP-1 cells has unveiled notable changes in specific genes (Fig 7A). Seeking a deeper understanding of the biological implications stemming from EF-24–induced gene dysregulation, we delved into IPA. Within the EF-24–treated THP-1

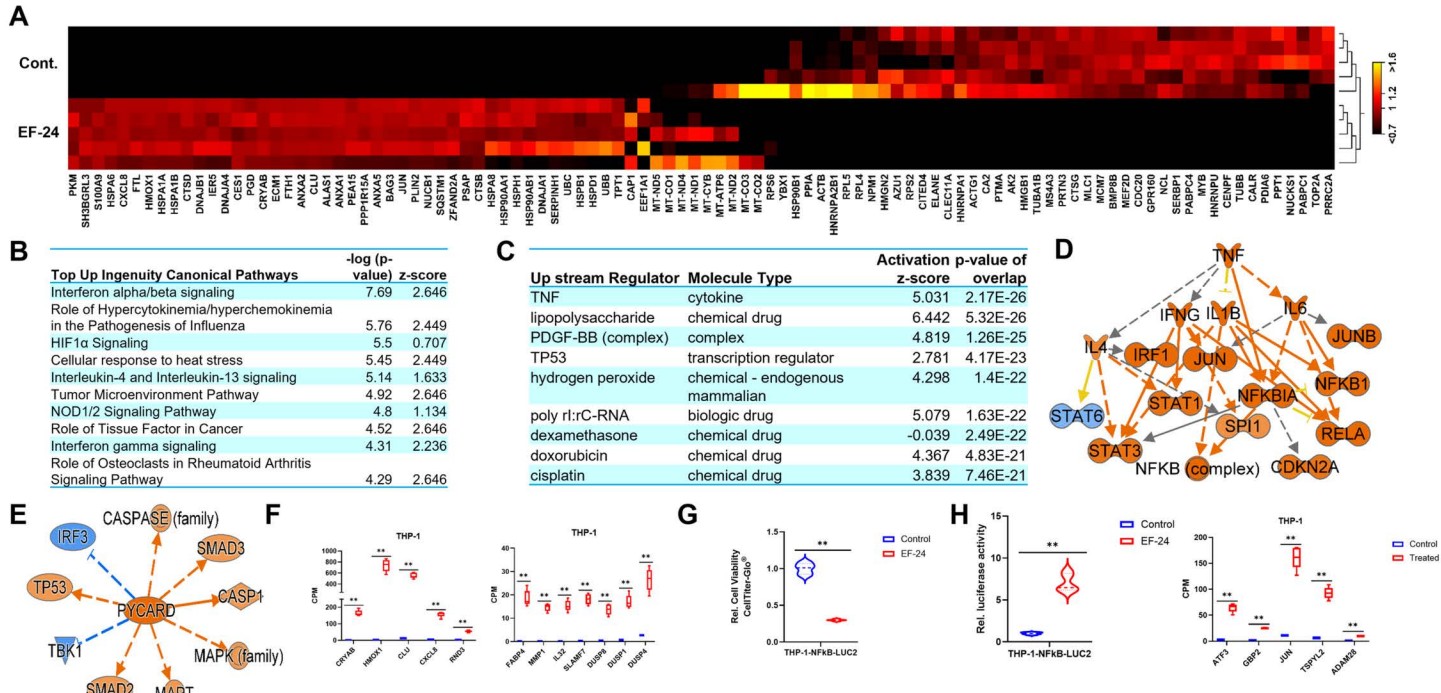

**Fig 7. EF-24 treatment modulates gene expression in THP-1 cells.** (A) Heatmap of the top 100 genes differentially expressed in EF-24–treated THP-1 cells (red = induced; black = reduced). (B, C) IPA revealed top canonical signaling pathways activated in EF-24–treated THP-1 cells. (D) TNF and (E) PYCARD activation observed in THP-1 cells treated with EF-24, functioning upstream of induced genes. (F) Box plots display quantitative levels of indicated genes in treated and untreated THP-1 cells. (G, H,) In EF-24–treated THP-1 cells, cell viability and NFκB activity show an inverse correlation. (G) Cell viability decreased in EF-24–treated THP-1 cells as compared to untreated control cells, whereas (H) NFκB-luciferase activity increased in EF-24–treated THP-1 cells as compared to untreated control cells. False Discovery Rate (FDR) p-value ** < 0.01.

cells, we observed the activation of interferon alpha/beta signaling—a pathway acknowledged for its role in inhibiting malignant cell growth through programmed cell death (Fig 7B) [40]. This underscores that the antitumorigenic effects of EF-24 in THP-1 cells are mediated by immune signaling. The genes upregulated and regulated by interferon signaling play pivotal roles in various biological processes and molecular functions.

Immunogenic cell death emerges as a significant mechanism in cancer therapy, harnessed by chemotherapy, radiation therapy, and targeted anticancer agents. This results in clinically relevant tumor-targeting immune responses [41] and culminates in immunogenic cell death (ICD) [42]. In our exploration of upstream regulators influencing genes altered in EF-24-treated THP-1 cells, TNF and P53 emerged as key orchestrators of these changes (Fig 7C,D). TNF, acting as a death ligand, binds to its cognate receptors, instigating complex I and complex II, which regulate caspase-8 activation. While complex I is associated with cell survival and proliferation, complex II executes cell death by activating caspase-3 and caspase-7 [42]. P53, a master regulator, suppresses tumorigenesis and surfaced as a prominent upstream regulator in EF-24–treated THP-1 cells [43]. P53 partially mediates the effects of EF-24 in inducing cell death in THP-1 cells.

Similarly, TNF regulatory networks displayed the activation of genes such as *IFNG*, *STAT1*, *JUN*, *CDKN2A*, *RALA*, and *NFKB1* in EF-24–treated THP-1 cells (Fig 7D). Activation of PYCARD resulted in downstream activation of genes such as *P53*, *MAPT*, *CASP1*, *SMAD2*, and *SMAD3* (Fig 7E). Subsequently, we identified candidate genes with differential expression and explored their biological associations; these genes exhibited a substantial increase in relative expression levels in EF-24–treated cells as compared to untreated THP-1 cells (Fig 7F). All these genes were found to be associated with cell death and survival, and their heightened activity contributed to induced cell death following EF-24 treatment.

To confirm TNF signaling activation in EF-24–treated cells, we performed a luciferase reporter assay using the THP-1-NFkB-LUC2 (ATCC® TIB-202-NFkB-LUC2™) cell line, which was derived from parental THP-1 cells by introducing NFkB-LUC2 genes to measure NFkB activity. The cells were subjected to EF-24 treatment or left untreated for 24 hours, after which we measured cell viability and NFkB luciferase activity. As depicted in earlier assays, cell viability significantly decreased in the EF-24–treated cells as compared to untreated cells (Fig 7G). In contrast, NFkB activity saw a significant multifold increase (Fig 7H). This further substantiates the transcriptomics results and establishes the activation of TNF signaling in EF-24–treated THP-1 cells.

## EF-24 differentially modulates p38/MAPK/ERK pathway targets in leukemia cell lines

In a previous study, Hsiao et al. demonstrated that EF-24 treatment activates the p38 mitogen-activated protein kinase (MAPK) pathway while attenuating ERK signaling in HL-60 acute myeloid leukemia (AML) cells after 24 hours [5]. To further investigate the transcriptional consequences of EF-24 exposure, we examined the expression of key downstream targets of the KEGG p38/MAPK pathway across several myeloid leukemia cell lines. In K-562 cells, EF-24T treatment led to significant upregulation of *STAT1*, *PAX6*, *MEF2D*, *ATF3*, *HSPB1*, *DDIT3* and *MEF2A*, while *MAX*, *RARA*, *HMGN1*, *CREB1*, and *RPS6KA5*, were downregulated (S2A Fig). In contrast, CCL-240 cells exhibited a general upward trend in gene expression, except for a moderate decrease in *MAPKAPK2*, *NFKB1*, *MAX*, *HMGN1*, and *ELK1* (S2B Fig), high-lighting a divergent response compared to K-562 cells. In Kasumi-1 cells, EF-24T induced MEF2D, *MAPKAPK2*, *ATF3, ATF2*, *CREB1*, *HSPBAP1*, *DDIT3*, *MAX*, *RPS6KA5, MEF2A*, *MEF2B*, and *HSPB1*, while suppressing STAT1, *NFKB1,* and ELK1 (S2C Fig). Finally, in THP-1 cells, EF-24T treatment resulted in increased expression of *STAT1*, *ETS1*, *RARA*, *ATF3*, *DDIT3*, and *MEF2A*, accompanied by reduced levels of *MYC*, *HMGN1*, *MEF2D*, *CREB1* and *RPS6KA5* (S2D Fig). We observed consistent upregulation of ATF3 and DDIT3 across all four cell lines, suggesting that EF-24 may exert part of its effect through these stress-responsive transcription factors downstream of p38/MAPK signaling. While several genes showed statistically significant changes following treatment, the overall magnitude of expression shifts was generally subtle to moderate. Moreover, only a limited subset of genes demonstrated consistent regulation across all cell lines. In addition to canonical KEGG p38/MAPK pathway target genes, Ingenuity Pathway Analysis identified a set of shared genes modulated by p38/MAPK activity across all four cell lines (S2E Fig). These findings underscore the context-dependent nature of EF-24's transcriptional impact, highlighting its selective modulation of p38/MAPK pathway targets involved in cel-lular stress responses and differentiation.

## Discussion

In this study, we examined the impact of EF-24 on the transcriptome of leukemia cell lines [44]. Despite numerous treat-ments available, myeloid leukemia – a disease with several genetically distinct subtypes – has seen few significant therapeutic advances [45]. EF-24 is known to inhibit growth and induce apoptosis in leukemia and other cancers via established molecular pathways; however, its effects on global transcription remain unknown [5,6,46–48]. To under-stand EF-24's molecular mechanisms, we used RNA sequencing to perform a whole transcriptome analysis on four well-established leukemia cell lines representing leukemia subtypes. Our goal was to identify gene expression signatures predictive of EF-24's effects on cancer cells for future studies. To ensure the reliability of our transcriptomic analysis, we employed a rigorous experimental design with a minimum of five biological replicates per condition, enhancing statistical power. Differentially expressed genes (DEGs) were identified using stringent criteria: a minimum transcript read count of 10–15 (approximately CPM ≥ 0.5) and an FDR-adjusted p-value < 0.05. Genes were retained only if they met these thresholds in at least one sample, reducing the likelihood of including transcripts with negligible or absent expression. Rather than preselecting genes for validation, we adopted a hypothesis-free, data-driven approach, prioritizing top-ranked DEGs with strong endogenous expressions in control samples. This strategy minimized the inclusion of low-abundance transcripts, which are more susceptible to technical noise and less likely to reflect biologically meaningful changes. The

consistency and magnitude of expression changes across replicates further support the robustness of our findings. We acknowledge, however, that this approach may have excluded low-expressing genes with potential biological relevance, particularly in mediating EF-24's effects. While such transcripts are more prone to background noise and off-target responses, and less likely to be directly linked to primary phenotypic outcomes (e.g., reduced cell viability), their exclusion represents a limitation of our study. Future investigations incorporating orthogonal validation methods may help elucidate the roles of these low-abundance genes.

We verified that EF-24 treatment exhibited potent antitumor activity, significantly reducing cell viability across all four treated myeloid leukemia cell lines that coincided with previous reports [5]. It was also noted that EF-24 treatment sensitivity is higher at lower concentrations in leukemia cell lines compared to solid tumor cell lines [49]. Transcriptomic analysis showed that EF-24's effects are mediated by a combination of genes and signaling molecules rather than a single pathway, highlighting its complex mechanism of action. The activation of pathways such as the S100 family signaling pathway [16], along with the identification of key molecules like IL6 and TNF, provided nuanced insights into the intricate molecular dynamics influenced by EF-24 [50,51]. Besides identifying the multifactorial genetic effects of EF-24 in selected myeloid leukemia cell lines, our analysis also confirmed the previously reported modulation of molecules and pathways in various cell lines from multiple tissue types treated with EF-24 [5,49]. In vitro, EF-24 significantly reduced the proliferation of various solid cancer cells while demonstrating non-cytotoxicity to normal cells. For instance, Zou et al. found that EF-24 inhibited the survival of gastric cancer cell lines SGC-7901 and BGC-823 but did not affect the survival of the normal human gastric epithelial cell line GES-1 or the rat kidney proximal tubular epithelial cell line NRK-52E [52]. In vivo, EF-24 demonstrated high oral bioavailability and low toxicity in mice, while effectively inhibiting the growth of human breast cancer in a mouse xenograft model [53]. Our findings explicitly underscore the EF-24's multifaceted nature in modulating genes associated with myeloid leukemia cell survival and proliferation.

In our analysis of K-562 cells, we uncovered a myriad of differentially regulated genes post-EF-24 treatment, shedding light on its mechanisms. Surprisingly, EF-24 treatment induced NFkB activity in K-562 cells, contrasting with previous reports of NFkB suppression in EF-24 treated lung, breast, ovarian, and cervical cancer cells [54]. Additionally, unlike the inhibition of HIF1α observed in certain cell lines, HIF1α was activated in myeloid leukemia cell lines upon EF-24 treatment [55,56]. Accordingly, EF-24-mediated inhibition of NFkB and HIF1α in cancer cells appears to be contextual, and its antitumorigenic effects are not simply exerted by regulating specific genes or pathways. However, we observed that the increased expression of transporter genes ABCB1, ABCB11, ABCA4, ABCA9, and many others did not contribute to the antiproliferative effects of EF-24, which is consistent with previous reports by Skoupa et al. [6]. Noteworthy inductions of apoptosis associated genes CLU and CRYAB, coupled with the activation of the wound healing signaling pathway unveiled EF-24's potential in orchestrating cellular responses [57,58]. This revelation emphasizes its prospective role in the context of chronic myeloid leukemia. Previous reports indicate that *Clusterin* (CLU) mediates apoptosis by interacting with Bcl-XL [59], aligning with the observed reduced cell viability in EF-24–treated cells. Attention was then directed to genes that exhibit higher levels of basal expression in untreated K-562 cells and overexpressed by EF-24, including *CLU*, *PTPRN*, *NDRG1*, *GBP2*, *OSGIN1*, *AFT3*, *IFI16*, *HLA-C*, *BEX2*, and *VWA5A*. Gene ontology analysis of the induced genes shed light on their potential antitumorigenic roles, suggesting tumor suppressor functions in EF-24–treated cells [60–68]. Notably, activation of wound healing pathway suggests a counter response to the acute cell-killing activity of EF-24 treatment in K-562 cells. Induction of cell death is inexorably linked with cancer therapy, but this can also initiate wound-healing processes that have been linked to cancer progression and therapeutic resistance [69,70]. Because the wound healing pathway was activated in EF-24–treated K-562 cells, understanding the therapeutic benefit of the curcumin analog EF-24 in chronic myeloid leukemia required further investigations. K-562 cells are known for their multipotentiality, exhibiting spontaneous differentiation into erythroid, granulocytic, and monocytic lineages. This capacity to activate alternative differentiation programs reflects characteristics of normal multipotent hematopoietic stem cells [71]. However, given their malignant origin, K-562 cells may not fully recapitulate the behavior of non-leukemic hematopoietic progenitors.

To more comprehensively understand the effects of EF-24 on hematopoietic signaling pathways, future studies should incorporate CD34 + normal hematopoietic stem and progenitor cells as a comparative reference. Such analyses would help delineate EF-24's impact on leukemic versus non-leukemic transcriptional programs and clarify whether its modulatory effects are lineage-specific or broadly applicable across hematopoietic contexts.

Shifting our focus to HL-60 cells, EF-24 treatment induced significant alterations in the expression of genes associated with cancer progression. The downregulation of the neutrophil degranulation pathway suggested an unfavorable environment for leukemia cell survival, while the concomitant upregulation of IL-10 signaling hinted at a complex interplay between proinflammatory, and cell survival pathways triggered by EF-24 [72]. Examining gene regulatory networks associated with molecular and cellular function, we observed the activation of leukocytes and differentiation of hematopoietic progenitor cells in EF-24–treated cells. While the activation of leukocytes is known to be part of the immune response, the impact on tumor burden requires further exploration in an in vivo setting. Additionally, the activation of genes associated with the differentiation of hematopoietic progenitor cells suggests that EF-24 exerts multifaceted antitumor effects in HL-60 cells [70].

Investigating upstream regulators of genes aberrantly activated in EF-24–treated cells, TNF emerged as a central regulator in HL-60 and THP-1 cells. TNF directly regulates upregulated genes, including *EGR1*, NFkB complex, *RELA*, and *IKBKB*, leading to downstream upregulation of *TP53*. EGR1 and TP53, known tumor suppressors, upregulate *p21*, inducing tumor cell apoptosis [73,74]. Glucocorticoid receptor (NR3C1) signaling, activated by dexamethasone, a known treatment for leukemia, was found to be negligible change, aligning with the observed negative control of genes regulated by dexamethasone in EF-24–treated HL-60 cells [75].

In Kasumi-1 cells, EF-24 induced a distinct shift toward tumor-suppressive mechanisms. The activation of TP53 and TP63, coupled with the downregulation of tumor promoters, pointed towards EF-24's potential to create a tumor-suppressive microenvironment. The modulation of key pathways, including the Pathogen Induced Cytokine Storm Signaling Pathway and IL-10 Signaling [40,72], added depth to our understanding of EF-24's impact on gene regulatory networks.

Finally, our investigation extended to THP-1 cells, revealing EF-24's prominent activation of the interferon alpha/beta signaling pathway [40]. This immune-mediated pathway, coupled with the involvement of TNF and P53 as upstream regulators, substantiated EF-24's potential to induce programmed cell death in malignant cells. Luciferase reporter assays further validated the activation of the TNF signaling cascade, reinforcing our transcriptomic findings.

It is also important to consider the dual roles that many signaling pathways play in cancer biology. Several of the pathways altered by EF-24 treatment—such as [insert specific pathways, e.g., NF-κB, PI3K/AKT, MAPK, IL-10, TNF]—are known to have context-dependent functions [76–80]. While these pathways often promote survival and proliferation in malignant cells, they can also trigger apoptotic or anti-proliferative responses under certain conditions. This complexity underscores the need for careful interpretation of pathway modulation and highlights the importance of validating these findings in diverse cellular and genetic contexts. Future studies will aim to dissect these dual roles more thoroughly, particularly in primary leukemia samples and across different lineage backgrounds.

While our study provides important insights into the effects of EF-24 on leukemia cell lines, we acknowledge several limitations that warrant further investigation. Notably, our current analysis was conducted using a limited panel of established leukemia cell lines—three AML and one CML—which, while informative, may not fully capture the heterogeneity and complexity of primary patient-derived leukemia samples. We have now explicitly acknowledged this limitation and emphasized the need for future studies involving primary leukemia cells to validate the translational relevance of our findings. Additionally, we recognize an imbalance in the number of AML versus CML cell lines used in this study. The inclusion of only a single CML cell line limits our ability to draw robust conclusions about lineage-specific responses to EF-24. While the study did not include FLT3-ITD mutated AML—a well-characterized and clinically significant subtype—the signaling principles elucidated here may still be relevant across broader AML contexts. Future work will expand the panel of CML cell lines, FLT-3-ITD mutated AML and other common subtypes and incorporate lymphoid leukemia models to enable a

more comprehensive comparison across hematologic lineages to assess the generalizability potential of the observed signaling dynamics. This will also allow for a more detailed investigation into whether EF-24 elicits differential responses based on leukemia lineage and maturation stage. Importantly, several keys signaling pathways were consistently altered by EF-24 treatment across the cell lines studied. These pathways, identified through transcriptomic analyses, will serve as focal points in subsequent studies aimed at delineating lineage-specific mechanisms of action. Such efforts will be critical for refining the therapeutic potential of EF-24 and identifying biomarkers predictive of response in diverse leukemia subtypes.

## Conclusion

Our comprehensive RNAseq analyses not only unraveled EF-24's potent antitumorigenic activity and its influence on transcriptional landscapes but also delved into its multifaceted impact on specific signaling pathways and gene networks controlling cell survival, proliferation, and immune responses in myeloid leukemia cells. These insights lay a solid foundation for the exploration of EF-24 as a promising therapeutic agent in myeloid leukemia treatment. Moving forward, further investigations in preclinical and clinical settings are imperative to harness the full potential of EF-24 for effective leukemia interventions.

## Materials and methods

### Cell lines

Preserved cell vials of the cell lines K-562 (ATCC° CCL-243™), HL-60 (ATCC° CCL-240™), Kasumi-1 (ATCC° CRL-2724™), and THP-1 (ATCC° TIB-202™) were acquired from the ATCC° repository (Manassas, VA, USA) and cultured according to specified parameters. These parameters were tailored for each cell line and adhered to the ATCC° cell culture laboratory, which operates under industry safety standards and quality guidelines and complies with ISO 9001 regulations. In brief, HL-60 cells were cultured in Iscove's Modified Dulbecco's Medium (IMDM; ATCC° 30–2005™) supplemented with fetal bovine serum (FBS; ATCC° 30–2020™) to a final concentration of 20%. Complete medium for K-562 consisted of IMDM with 10% FBS. For Kasumi-1, complete medium was prepared using RPMI-1640 (ATCC° 30–2001™) supplemented with 20% FBS. THP-1 cells were cultured in RPMI-1640 supplemented with 10% FBS. All cell lines were maintained at 37°C in an atmosphere of 95% air and 5% $CO_2$.

### Chemical and reagents

EF-24 (with a purity of ≥98%; determined by HPLC (High Performance Liquid Chromatography) was procured from Sigma-Aldrich° (St. Louis, MO, USA; catalog number E8409-25MG). It was dissolved in dimethyl sulfoxide (DMSO; ATCC° 4-X™) at a concentration of 25 mg (milligram) EF-24 per 2.5 mL (milliliter) DMSO, yielding a stock concentration of 32.1203 mM (millimolar) according to the manufacturer's instructions. Subsequently, a 10 mM solution was prepared from the stock by combining 312.5 µL (microliter) of the 32 mM EF-24 solution with 687.5 µL of DMSO to achieve a final volume of 1 mL. The final treatment concentration was 2 µM; for instance, 10 µL of the 10 mM EF-24 solution was added to 50 mL of media to attain a final concentration of 2 µM in cell culture.

### Cytotoxicity assay

The cytotoxic effect of EF-24 on AML cell lines was assessed using the CellTiter-Glo° Luminescent Cell Viability Assay kit (Promega° catalog number G7570). K-562, HL-60, Kasumi-1, and THP-1 were seeded into T25 flasks (Corning, part no. 430639) at a density of $3 \times 10^6$ cells per flask. Subsequently, the cells were treated with a concentration of 2 µM EF-24 for 24 hours. Following treatment, 100 µL of cell suspension was combined with 100 µL of CellTiter-Glo° solution in a 96-well plate with each well containing $3 \times 10^4$ cells. The plate was then incubated at room temperature for 10 minutes, after

which the absorbance was measured at 450 nm using a microplate reader (Molecular Devices SpectraMax®). Each experiment was performed twice with a minimum of five biological replicates.

## RNA extraction and quality control (QC)

RNA isolation was performed using the QIAGEN® QIAcube® automated system with the RNeasy Mini QIAcube® Kit. Frozen samples were thawed and prepared for RNA extraction according to ATCC's work instructions. Extracted samples were tested for RNA integrity and quality using the Agilent® TapeStation™ (RNA Integrity Number (RIN) ≥ 6.5), RNA purity using the Thermofisher® Nanodrop™ (A260/A280 1.8 ≥ x ≤ 2.2), and concentration using the Qubit®™.

## RNA-Seq library preparation and sequencing

Automated RNA-seq NGS library preparation was performed on the Eppendorf epMotion® 5075 Liquid Handler using the Illumina® Stranded mRNA Prep, Ligation kit. Prepared NGS libraries were assessed by quantitative analysis using the Invitrogen®™ Qubit®™ dsDNA High Sensitivity Assay Kit and qualitative analysis using the Agilent® 4200 TapeStation®™ and D5000 ScreenTape System. Libraries were loaded on an Illumina® P3 200-cycle Reagent kit and sequenced on the NextSeq® 2000 platform.

## RNA-Seq and bioinformatics analysis

Our data analysis pipeline included quality control, read trimming, alignment to the reference transcriptome, and quantification of gene expression. Utilizing the CLC Genomics Workbench v23 (QIAGEN Digital Insights), an end-to-end pipeline was created that, briefly, entailed the following steps. First, raw paired end Illumina reads were trimmed and filtered to a minimum quality of Q30 and a maximum of 2 ambiguous bases. Furthermore, potential "read-through" adapter sequences and 3' polyG sequences (due to using the 2-color NextSeq platform) were also automatically identified and trimmed. Reads below 50 bp were then discarded. Next, reads were mapped to the human genome hg38 reference genome (obtained from Ensemble) using the default settings for bulk-RNAseq experiments: mismatch cost 2; InDel cost 3; length fraction 0.8; similarity fraction 0.8; maximum hits per reads 10; reversed strand-specific mapping 1; ignore broken read pairs 1. A minimum of 18M mapped reads per library were required for each biological replicate, and TMM normalization was carried out for each library (Robinson and Oshlack, 2010). Statistical comparisons between groups were conducted using a Wald test, and fold-change values calculated from the GLM model (Robinson et al., 2010). Outliers were down weighted and iteratively re-fit to the GLM model (Zhou et al., 2014). Low expression genes were filtered prior to FDR correction and calculation (Love et al., 2014). Count-per-million (CPM) values are used to filter genes in RNA-seq analysis. A CPM value of 0.5 is usually equivalent to 10–15 reads. A gene is typically retained if it has a CPM above 0.5 in at least one sample, as a smaller count indicates that the gene is not expressed in that sample. The gene list obtained from the differential gene expression analysis was further analyzed using the Ingenuity Pathway Analysis (IPA) platform (Krämer et al., 2014). IPA automatically carried out statistical comparisons for gene-set enrichment analysis, statistically significant pathway activation, identification of upstream regulators and downstream targets, gene and pathway associated diseases and biofunctions. IPA was also used to produce network diagrams of pathways and associated differentially expressed genes.

## Supporting information

**S1 Fig. Expression analysis of downstream target genes in the p38/MAPK signaling pathway across myeloid leukemia cell lines.** (A–D) Bar plots show the relative mRNA expression levels of the indicated genes in EF-24–treated versus untreated control samples for each cell line representing distinct myeloid leukemia subtypes. False Discovery Rate (FDR) p-value ** < 0.01.
(TIF)

**S2 Fig.  Expression analysis of downstream target genes in the IL-10 signaling pathway across myeloid leukemia cell lines.** (A–C) Bar plots show the relative mRNA expression levels of the indicated genes in EF-24–treated versus untreated control samples for each cell line representing distinct myeloid leukemia subtypes. False Discovery Rate (FDR) p-value **<0.01.
(TIF)

## Author contributions

**Conceptualization:** Ajeet P. Singh, Jonathan L. Jacobs.

**Data curation:** Jonathan L. Jacobs.

**Formal analysis:** Ajeet P. Singh, Jonathan L. Jacobs.

**Investigation:** Ajeet P. Singh, Noah Wax, James Duncan, Ana S. Fernandes.

**Methodology:** Ajeet P. Singh.

**Project administration:** Jonathan L. Jacobs.

**Supervision:** Ana S. Fernandes, Jonathan L. Jacobs.

**Visualization:** Ajeet P. Singh.

**Writing – original draft:** Ajeet P. Singh.

**Writing – review & editing:** Ajeet P. Singh, Noah Wax, James Duncan, Ana S. Fernandes, Jonathan L. Jacobs.

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
