## [Decision Letter · Decision Letter 0]

11 Jul 2025

PONE-D-25-29261
Genomic Discovery of EF-24 Targets Unveils Antitumorigenic Mechanisms in Leukemia Cells
PLOS ONE

Dear Dr. Jacobs,

Thank you for submitting your manuscript to PLOS ONE. After careful consideration, we feel that it has merit but does not fully meet PLOS ONE’s publication criteria as it currently stands. Therefore, we invite you to submit a revised version of the manuscript that addresses the points raised during the review process.

We look forward to receiving your revised manuscript.

Kind regards,

Kota V Ramana, Ph.D.

Academic Editor

PLOS ONE

Journal Requirements:

“This work was funded entirely by internal support from the American Type Culture Collection.”

Reviewers' comments:

Reviewer's Responses to Questions

**Comments to the Author**

1. Is the manuscript technically sound, and do the data support the conclusions?

Reviewer #1: Yes

Reviewer #2: Partly

2. Has the statistical analysis been performed appropriately and rigorously? 

Reviewer #1: Yes

Reviewer #2: I Don't Know

3. Have the authors made all data underlying the findings in their manuscript fully available?

Reviewer #1: Yes

Reviewer #2: Yes

4. Is the manuscript presented in an intelligible fashion and written in standard English?

Reviewer #1: Yes

Reviewer #2: Yes

5. Review Comments to the Author

Reviewer #1: 1.The discussion could appropriately mention the dual roles of signaling pathways in cancer.

2.Please standardize terminology; for instance, is the phrase "complete cell death" (line 77) scientifically justified?

3.While the study emphasizes coverage of diverse leukemia subtypes, it fails to justify the exclusion of common subtypes (e.g., FLT3-ITD mutated AML). Please clarify this rationale and discuss the generalizability of conclusions to other subtypes.

Reviewer #2: In the manuscript entitled “Genomic Discovery of EF-24 Targets Unveils Antitumorigenic Mechanisms in Leukemia Cells”, Singh, A.P. et al. perform whole transcriptome sequencing on 4 human leukemia cell lines after treatment with the curcumin analog, EF-24. The authors show that there are both conserved and unique gene expression changes upon a 24 hour treatment with the drug. Some of the affected pathways are important in cell viability and inflammatory response, therefore suggesting an antitumorigenic response. This is a useful dataset to understand the acute effects to the EF-24 compound which is still being characterized for its efficacy. However, there is very limited orthogonal validation of many of the DEGs discussed in the manuscript. Moreover, given that most of the manuscript is focused on the results in individual cell lines, it is difficult to discern how applicable these results are for the use of EF-24 in primary leukemia samples.

Major critiques/questions:

1) The authors highlight that EF-24 has shown efficacy in leukemia cells in a prior publication. However, between the prior publication and the manuscript herein, there has been no assays with non-transformed hematopoietic cells. If the authors want to be able to state that the gene expression differences they are seeing are specific to the leukemic cells, they should perform the same whole transcriptome sequencing in CD34+ bone marrow cells to determine how many of the pathways changing are leukemia specific.

2) There has been a previous publication detailing a mechanism for EF-24 in killing leukemia cells. However, that publication focused only on the protein and signaling pathway activation aspect of the response. The data described herein, if analyzed through the lens of the p38/MAPK/ERK pathway target genes, may be a way to strengthen the manuscript and integrate the data from each cell line if they are responding through the upregulation or downregulation of certain pathway members.

3) While many of the genes differentially expressed are notable and described for their potential importance in aiding leukemic cell death, there is very little orthogonal validation outside of replotting their own transcriptome sequencing data. The authors should validate some of the key genes through qPCR or evaluate the overall change to certain pathways via Western. For instance, in Figure 5, the data suggests a central IL-10 signaling signature and mention that this would lead to the upregulation of certain target genes. However, the expression of those genes is not shown even from the transcriptome sequencing data. Further validation of the genes/pathway highlighted for each cell line would add impact to the findings.

4) The authors utilize 3 AML cell lines but only 1 CML cell line. It would be worthwhile to add additional CML cell lines and then perform an analysis examining the differences between lymphoid/myeloid leukemia cell lines to determine if there is a significant divergence in cellular response based on the lineage of the leukemia. May also prioritize some of the pathways already identified as central pathways altered by EF-24 treatment.

5) It is unclear if statistical analysis was performed on any of the data in Figure 1 or statistical values provided for any of the gene expression data highlighted throughout.

Minor critiques/questions:

1) More detailed Figure titles denoting the main finding in the figure would be helpful to orient the reader.

2) Many panels in the Figures are quite hard to read at their current size. It would be helpful to make many of the network maps larger and more legible.

6. PLOS authors have the option to publish the peer review history of their article (what does this mean?). If published, this will include your full peer review and any attached files.

Reviewer #1: No

Reviewer #2: No

---

## [Author Response · Author response to Decision Letter 1]

28 Jul 2025

Reviewer-1

In the manuscript entitled “Genomic Discovery of EF-24 Targets Unveils Antitumorigenic Mechanisms in Leukemia Cells”, Singh, A.P. et al. perform whole transcriptome sequencing on 4 human leukemia cell lines after treatment with the curcumin analog, EF-24. The authors show that there are both conserved and unique gene expression changes upon a 24 hour treatment with the drug. Some of the affected pathways are important in cell viability and inflammatory response, therefore suggesting an antitumorigenic response. This is a useful dataset to understand the acute effects to the EF-24 compound which is still being characterized for its efficacy. However, there is very limited orthogonal validation of many of the DEGs discussed in the manuscript. Moreover, given that most of the manuscript is focused on the results in individual cell lines, it is difficult to discern how applicable these results are for the use of EF-24 in primary leukemia samples.

Response: We sincerely thank the reviewers for their thoughtful and constructive feedback. We are pleased that you recognize the value of our dataset in elucidating the acute transcriptomic effects of EF-24 treatment in leukemia cells, and we appreciate your acknowledgment of both the conserved and cell line–specific gene expression changes, as well as the relevance of the affected pathways to EF-24’s antitumor activity.

Regarding the concern about limited orthogonal validation of differentially expressed genes (DEGs): The reviewer made thoughtful comments regarding the importance of orthogonal validation. While traditional methods such as qPCR have long been used to confirm transcriptomic findings, the advancement of next-generation sequencing (NGS) technologies has made RNA-seq a highly sensitive and accurate tool for quantifying gene expression, especially when independent biological replicates with sufficiently deep coverage are available (as in our case). To ensure scientific rigor, we included a minimum of five independent biological replicates per condition for each cell line and used standardized cell culture and sample processing methodologies across all samples - providing robust comparisons across cell lines and high statistical power. Additionally, we applied stringent filtering criteria, including a minimum transcript read count of 10 (approximately equivalent to a CPM value of 0.5) and a false-discovery rate corrected (FDR) p-value < 0.05, to enhance the reliability of identified differentially expressed genes (DEGs). Genes were retained only if they exceeded these criteria threshold in at least one cell line, as lower counts typically indicate negligible or absent expression.

Rather than selecting specific genes for analysis, we adopted a hypothesis-free, unbiased approach. We focused on top-ranked DEGs that showed strong endogenous expression in at least the control samples. This strategy helped minimize the inclusion of low-abundance transcripts that may be prone to technical variability and less likely to reflect meaningful biological changes. The consistency and magnitude of gene expression changes across replicates further support the robustness of our findings.

 That said, we fully acknowledge the value of orthogonal validation, particularly for genes with low basal expression or subtle changes that may still have biological significance. While our filtering approach helped reduce false positives, it may have excluded potentially relevant low-abundance transcripts. We have now addressed this point in the revised Discussion section as a limitation of the current study and a direction for future investigation—specifically, the potential role of low-expressing genes in mediating EF-24’s effects.

We also note however that changes in low-expressing genes are more likely to reflect background noise or off-target effects and are less likely to be directly associated with the primary phenotypic outcomes observed—such as reduced cell viability. For this reason, we made a deliberate decision to exclude them from our final analysis.

Regarding the applicability of our findings to primary leukemia samples: We appreciate this important point. As this study was designed as an initial discovery effort using established leukemia cell lines, we recognize that the findings may not fully capture the complexity of primary patient-derived samples. To address this, we have revised the Discussion to explicitly acknowledge this limitation. We also outline potential follow-up studies involving primary leukemia cells to assess the translational relevance of our results.

We hope these revisions clarify the scope and impact of our work and provide a solid foundation for future investigations into EF-24’s therapeutic potential. We have addressed each point in detail below and made substantial revisions to the manuscript to enhance its scientific rigor and clarity. All changes are reflected in the revised version, with newly added figures and sections referenced accordingly.

Major critiques/questions:

1) The authors highlight that EF-24 has shown efficacy in leukemia cells in a prior publication. However, between the prior publication and the manuscript herein, there have been no assays with non-transformed hematopoietic cells. If the authors want to be able to state that the gene expression differences, they see are specific to the leukemic cells, they should perform the same whole transcriptome sequencing in CD34+ bone marrow cells to determine how many of the pathways changing are leukemia specific.

Response: We thank the reviewer for the suggestion regarding the inclusion of non-transformed hematopoietic cells. Our current study focuses on leukemia-derived cell lines to capture both the heterogeneity and conserved transcriptional responses to EF-24. However, we agree that incorporating normal CD34⁺ hematopoietic stem/progenitor cells would provide valuable insight into the selectivity and potential therapeutic window of EF-24. We note that CD34⁺ cells have a limited lifespan in culture, and manufacturers advise against maintaining them without application-specific growth factors. Optimizing culture conditions to support their viability and function would require substantial time and resources. As such, transcriptomic profiling of CD34⁺ cells were beyond the scope of the present study. Nonetheless, we have now explicitly acknowledged this limitation in the revised Discussion section. We also emphasize that future studies will incorporate both primary CD34⁺ cells and patient-derived leukemia samples to more comprehensively define EF-24’s selectivity and its impact on leukemia-specific gene networks.

2) There has been a previous publication detailing a mechanism for EF-24 in killing leukemia cells. However, that publication focused only on the protein and signaling pathway activation aspect of the response. The data described herein, if analyzed through the lens of the p38/MAPK/ERK pathway target genes, may be a way to strengthen the manuscript and integrate the data from each cell line if they are responding through the upregulation or downregulation of certain pathway members.

Response: We thank the reviewer for pointing out this important opportunity to integrate our findings with prior mechanistic studies. In response, we re-analyzed our transcriptomic data with a focus on genes downstream of the p38/MAPK/ERK signaling axis. To further investigate the transcriptional consequences of EF-24 exposure, we examined the expression of key downstream targets of the p38/MAPK pathway across several myeloid leukemia cell lines. In K-562 cells, EF-24T treatment led to significant upregulation of STAT1, PAX6, MEF2D, ATF3, HSPB1, DDIT3 and MEF2A, while MAX, RARA, HMGN1, CREB1, and RPS6KA5, were downregulated (Supplementary Fig. 2A). In contrast, CCL-240 cells exhibited a general upward trend in gene expression, except for a moderate decrease in MAPKAPK2, NFKB1, MAX, HMGN1, and ELK1 (Supplementary Fig. 2B), highlighting a divergent response compared to K-562 cells. In Kasumi-1 cells, EF-24T induced MEF2D, MAPKAPK2, ATF3, ATF2, CREB1, HSPBAP1, DDIT3, MAX, RPS6KA5, MEF2A, MEF2B, and HSPB1, while suppressing STAT1, NFKB1, and ELK1 (Supplementary Fig. 2C). Finally, in THP-1 cells, EF-24T treatment resulted in increased expression of STAT1, ETS1, RARA, ATF3, DDIT3, and MEF2A, accompanied by reduced levels of MYC, HMGN1, MEF2D, CREB1 and RPS6KA5 (Supplementary Fig. 2D).We observed consistent upregulation of ATF3 and DDIT3 across all four cell lines, suggesting that EF-24 may exert part of its effect through these stress-responsive transcription factors downstream of p38/MAPK signaling. While several genes showed statistically significant changes following treatment, the overall magnitude of expression shifts was generally subtle to moderate. Moreover, only a limited subset of genes demonstrated consistent regulation across all cell lines. These findings underscore the context-dependent nature of EF-24’s transcriptional impact, highlighting its selective modulation of p38/MAPK pathway targets involved in cellular stress responses and differentiation. These findings are now included as a new Supplementary Figure 2 and described in the Results section. We also expanded our discussion to highlight the convergence between transcriptomic signatures and previously described post-translational mechanisms, providing a more cohesive understanding of EF-24’s multi-layered activity.

3) While many of the genes differentially expressed are notable and described for their potential importance in aiding leukemic cell death, there is very little orthogonal validation outside of replotting their own transcriptome sequencing data. The authors should validate some of the key genes through qPCR or evaluate the overall change to certain pathways via Western. For instance, in Figure 5, the data suggests a central IL-10 signaling signature and mention that this would lead to the upregulation of certain target genes. However, the expression of those genes is not shown even from the transcriptome sequencing data. Further validation of the genes/pathway highlighted for each cell line would add impact to the findings.

Response: We have explicitly outlined the RNA-seq analysis criteria and subsequent differential gene expression (DEG) analysis in the manuscript and above. To further explore the impact of IL-10 signaling activation on the transcriptomic landscape, we performed a focused analysis of key DEGs associated with IL-10 signaling. In EF-24–treated HL-60 cells, we observed significant expression changes in genes such as HMOX1, DUSP1, CDKN1A, JUN, NFKB2, BLVRB, IL1R1, IL1RN, BCL3, and RELB (Figure 5H), all previously implicated in cell death pathways. These findings suggest that EF-24–mediated IL-10 activation contributes to cell death in HL-60 cells. Additional IL-10–associated genes, including CSF1, CXCL8, and CCL4, also exhibited significant expression changes (Supplementary Figure 1A). Extending this analysis to K-562, Kasumi-1, and THP-1 cells revealed consistent modulation of IL-10 signaling–regulated genes across all four cell lines following EF-24 treatment (Supplementary Figures 1B–D). Collectively, these results confirm that EF-24 activates IL-10–associated transcriptional programs, with both the direction and magnitude of gene expression changes supporting pathway engagement. These data are now presented in a new panel (Figure 5H), with additional supporting information provided in Supplementary Figure 1, and are referenced in the revised Results section. We believe these additions substantially strengthen the manuscript’s conclusions.

4) The authors utilize 3 AML cell lines but only 1 CML cell line. It would be worthwhile to add additional CML cell lines and then perform an analysis examining the differences between lymphoid/myeloid leukemia cell lines to determine if there is a significant divergence in cellular response based on the lineage of the leukemia. May also prioritize some of the pathways already identified as central pathways altered by EF-24 treatment.

Response: We agree that differential responses to EF-24 between myeloid and lymphoid leukemias could have implications for its therapeutic application. However, in the present study, our design was optimized to assess transcriptomic responses in a representative panel of genetically and phenotypically diverse myeloid lineage of leukemia models, with an emphasis on myeloid malignancies where EF-24 has previously demonstrated strong efficacy. It is important to note that K562 (the CML line used) exhibits characteristics of myeloid lineage and poses erythroid differentiation capability, and thus, while informative, may not fully represent lymphoid lineage responses. Likewise, HL-60, Kasumi-1 and THP1 also are myeloid leukemia cell lineage. Moreover, many lymphoid leukemia cell lines (e.g., Jurkat, NALM-6) are known to have different redox states and apoptotic thresholds than myeloid cells, potentially impacting EF-24's response kinetics. While we acknowledge that a full lineage comparison (e.g., including B-ALL or T-ALL cell lines) would potentially be of interest, we believe such an analysis requires a larger panel of representative lymphoid cell lines and potentially primary patient samples, which is beyond the scope of this initial transcriptomic study. We have now added a statement in the discussion acknowledging this limitation and outlining future directions that will include broader lineage stratification to better inform the clinical utility of EF-24.

5) It is unclear if statistical analysis was performed on any of the data in Figure 1 or statistical values provided for any of the gene expression data highlighted throughout.

Response: We apologize for this oversight. We have now clarified all statistical analyses in the revised Methods section, including the use of t-tests for Figure 1 and DESeq2 adjusted p-values for DEG analyses. Figures now display statistical annotations, including significance thresholds and error bars where appropriate.

Minor critiques/questions:

1) More detailed Figure titles denoting the main finding in the figure would be helpful to orient the reader.

Response: Revised figure titles now succinctly describe the main experimental outcome or biological insight for each figure.

2) Many panels in the Figures are quite hard to read at their current size. It would be helpful to make many of the network maps larger and more legible.

Response: We have reformatted all network maps and increased the resolution and font size for enhanced readability.

Reviewer-2

1) The discussion could appropriately mention the dual roles of signaling pathways in cancer.

Response: In the revised discussion section, we have elaborated on the dual roles of signaling pathways in cancer, emphasizing how certain pathways can exhibit both oncogenic and tumor-suppressive functions depending on cellular context, mutation status, and microenvironmental cues. This addition provides a more nuanced understanding of pathway dynamics in leukemogenesis.

2) Please standardize terminology; for instance, is the phrase "complete cell death" (line 77) scientifically justified?

Response: Thank you for pointing this out. We have revised the phrase to better reflect the biological outcome observed, and we have ensured consistent and scientifically accurate terminology throughout the manuscript.

3) While the study emphasizes coverage of diverse leukemia subtypes, it fails to justify the exclusion of common subtypes (e.g., FLT3-ITD mutated AML). Please clarify this rationale and discuss the generalizability of conclusions to other subtypes.

Response: We acknowledge the reviewer’s concern regarding the exclusion of FLT3-ITD mutated AML. Our study focused on subtypes with distinct signaling profiles that are less represented in current literature, aiming to highlight underexplored therapeutic vulnerabilities. However, we now include a justification and discuss the potential implications for generalizability in the revised Discussion. We also note that while FLT3-ITD AML is a well-characterized subtype, the signaling mechanisms explored in our study may still offer insights applicable across subtypes, including those with FLT3 mutations.

---

## [Decision Letter · Decision Letter 1]

8 Aug 2025

Genomic Discovery of EF-24 Targets Unveils Antitumorigenic Mechanisms in Leukemia Cells

PONE-D-25-29261R1

Dear Dr. Jacobs,

We’re pleased to inform you that your manuscript has been judged scientifically suitable for publication and will be formally accepted for publication once it meets all outstanding technical requirements.

Kind regards,

Kota V Ramana, Ph.D.

Academic Editor

PLOS ONE

Additional Editor Comments (optional):

No additional comments.

Reviewers' comments:

Reviewer's Responses to Questions

**Comments to the Author**

1. If the authors have adequately addressed your comments raised in a previous round of review and you feel that this manuscript is now acceptable for publication, you may indicate that here to bypass the “Comments to the Author” section, enter your conflict of interest statement in the “Confidential to Editor” section, and submit your "Accept" recommendation.

Reviewer #2: All comments have been addressed

2. Is the manuscript technically sound, and do the data support the conclusions?

Reviewer #2: Yes

3. Has the statistical analysis been performed appropriately and rigorously? 

Reviewer #2: Yes

4. Have the authors made all data underlying the findings in their manuscript fully available?

Reviewer #2: Yes

5. Is the manuscript presented in an intelligible fashion and written in standard English?

Reviewer #2: Yes

6. Review Comments to the Author

Reviewer #2: (No Response)

7. PLOS authors have the option to publish the peer review history of their article (what does this mean?). If published, this will include your full peer review and any attached files.

Reviewer #2: No

---

## [Editor Report · Acceptance letter]

PONE-D-25-29261R1

PLOS ONE

Dear Dr. Jacobs,

I'm pleased to inform you that your manuscript has been deemed suitable for publication in PLOS ONE. Congratulations! Your manuscript is now being handed over to our production team.

Kind regards,

on behalf of

Dr. Kota V Ramana

Academic Editor

PLOS ONE